# The Spectrum of Interactions between *Cryptococcus neoformans* and Bacteria

**DOI:** 10.3390/jof5020031

**Published:** 2019-04-12

**Authors:** François L. Mayer, James W. Kronstad

**Affiliations:** 1Michael Smith Laboratories, University of British Columbia, Vancouver, BC V6T 1Z4, Canada; fmayer@msl.ubc.ca; 2Department of Microbiology and Immunology, University of British Columbia, Vancouver, BC V6T 1Z4, Canada

**Keywords:** *Cryptococcus*, fungal pathogen, bacteria, interkingdom-interactions, antifungal activity

## Abstract

*Cryptococcus neoformans* is a major fungal pathogen that infects immunocompromised people and causes life-threatening meningoencephalitis. *C. neoformans* does not occur in isolation either in the environment or in the human host, but is surrounded by other microorganisms. Bacteria are ubiquitously distributed in nature, including soil, and make up the dominant part of the human microbiota. Pioneering studies in the 1950s demonstrated antifungal activity of environmental bacteria against *C. neoformans*. However, the mechanisms and implications of these interactions remain largely unknown. Recently, interest in polymicrobial interaction studies has been reignited by the development of improved sequencing methodologies, and by the realization that such interactions may have a huge impact on ecology and human health. In this review, we summarize our current understanding of the interaction of bacteria with *C. neoformans*.

## 1. Introduction

Most microorganisms on earth do not occur in isolation, but interact with other microbes. This is true for interactions among members of the same kingdom, such as between different species of bacteria, as well as for mixed populations of multi-kingdom microbial consortia, e.g., interactions between bacteria and fungi [1,2]. Bacterial interactions with fungal pathogens are not only biologically interesting but may also provide new opportunities for antifungal therapy. Fortunately, the vast majority of fungi are harmless for humans [3]. However, a small number of fungi are human pathogens and can cause life-threatening infections [4]. Some of the predominant fungal pathogens include the human-associated fungus *Candida albicans*, and the environmental fungi *Cryptococcus neoformans* and *Aspergillus fumigatus* [5]. Fungal infections are usually very difficult to treat with current antifungal drugs, and the incidence of worldwide fungal resistance is on the rise [6].

*C. neoformans* is a common human fungal pathogen, and the yeast is responsible for almost a quarter million deaths annually [7]. The fungus primarily affects people with impaired immune systems, especially patients suffering from HIV/AIDS. Indeed, *C. neoformans* is estimated to be responsible for a staggering 15% of all AIDS-associated deaths worldwide [7]. Natural habitats for *C. neoformans* include soil, trees, and bird excreta [8,9,10]. *C. neoformans* cells are thought to be present in different morphologies in nature. Fungal cells can occur as desiccated yeast cells or as spores, which have the tendency to be distributed by wind and animals. For example, it has been shown that *C. neoformans* can be found attached to the feet and beaks of pigeons [11]. Interestingly, birds are not susceptible to the fungus. Humans, however, can inhale *C. neoformans* spores and yeast cells, leading to the development of pulmonary infections [12]. Such infections of the lung can further develop into life-threatening cryptococcal meningitis in a process involving the transition of the fungus from the lung to the brain via crossing the blood brain barrier.

So far, studies of fungal–bacterial interaction have mainly been performed with *C. albicans*, and it was found that these interkingdom interactions can dramatically influence human health and disease [13,14,15,16,17]. Relatively few studies have addressed interkingdom interactions between *C. neoformans* and bacteria. As mentioned, *C. neoformans* occurs both in the environment, for example in soil, and inside humans during infection, in comparison to *C. albicans*—which is obligately associated with a mammalian host. Therefore, *C. neoformans* is likely to come into contact and potentially interact with an enormous number of bacteria in its natural habitats and with the human microbiota during disease [18,19]. In this review, we provide background context and highlight new studies indicating that specific bacteria can have dramatic effects on cryptococcal growth and virulence factor expression. Understanding such interactions may lead to the discovery of novel antifungal drug targets or novel antifungal drugs.

## 2. Types of Fungal–Bacterial Interactions

There are several ways that fungi and bacteria can interact with each other. Considering a bi-microbial interaction between a fungus and a bacterium, the outcome can be negative, positive, or neutral for both, or for either partner. In total, this results in nine possible interaction outcomes (Figure 1a). A useful illustrative example of positive and negative interactions comes from the bacterial pathogen *Pseudomonas aeruginosa* and *C. albicans,* which often co-infect patients with cystic fibrosis (CF). *P. aeruginosa* forms recalcitrant biofilms with increased antibiotic tolerance in the lungs, and it was shown that biofilm formation is promoted by *C. albicans* through ethanol production by the fungus [20]. *P. aeruginosa* itself stimulates fungal ethanol production by generating redox-active small molecules termed phenazines. This cyclic relationship was proposed to establish favorable conditions for both *P. aeruginosa* and *C. albicans* to co-infect CF patients [20]. Indeed, co-infection results in a significantly worse clinical outcome for CF patients compared with infection with *P. aeruginosa* alone [21]. The interaction between *P. aeruginosa* and *C. albicans* is more complex, however, and can have different outcomes than those mentioned in the context of CF. In general, pseudomonal phenazines are toxic to *C. albicans* at higher concentrations. At lower concentrations, phenazines were demonstrated to inhibit key *C. albicans* virulence factors, for example the yeast-to-hypha transition, adhesion to surfaces, and biofilm formation [22,23]. Conversely, in the context of gastrointestinal infections, *C. albicans* was recently shown to produce factors that inhibit *P. aeruginosa* virulence by suppressing production of the siderophores pyochelin and pyoverdine [24]. Notably, this process did not impact bacterial growth or gut colonization.

For *C. neoformans*, interactions with bacteria could have a positive outcome for the fungus in the form of enhanced proliferation, or stimulation of protective fungal virulence factors such as melanin and capsule formation (Figure 1b). On the other hand, bacterial activities that result in inhibition of these processes may confer negative outcomes for *C. neoformans* (Figure 1b). Fungal proliferation is clearly important in competing for limited space and nutrient sources in a given niche. As virulence factors, melanin and capsule provide readily assayable readouts to observe the impact of bacteria. Melanin is a dark-brown/black, cell wall-anchored pigment that protects *C. neoformans* from multiple stresses in the environment and during host infection [25,26,27,28]. Melanin formation is one of the main cryptococcal virulence factors, and mutants with defects in melanization are usually attenuated for virulence in vivo [29,30,31]. The most important virulence factor in *C. neoformans*, however, is the polysaccharide capsule [32,33,34]. Polysaccharide fibers are anchored to the α-1,3-glucan layer of the cell wall and protect cells from phagocytosis by amoeba in the environment or macrophages in the human host [12,35]. Moreover, capsule polysaccharide can modulate the human immune response [36]. Capsule-deficient mutants are usually avirulent or strongly reduced for virulence in animal models [30,37,38,39]. Below we discuss the impact of specific bacterial species on cryptococcal proliferation and virulence factor production.

## 3. Interaction of *C. neoformans* with Bacteria

Cells of *C. neoformans* can be found in pigeon excreta, suggesting that the pigeon gastrointestinal tract is at least temporarily colonized with this fungus. Interestingly, pigeons as well as other birds are not susceptible to cryptococcal disease, and fungal cells are completely cleared from the excreta within 4 weeks [40]. This observation suggests that factors within the gastrointestinal tract may have fungicidal activity. In an attempt to examine the impact of the natural pigeon gut microbiota, researchers in the late 1970s isolated bacteria from the intestinal contents of healthy pigeons and studied their impact on *C. neoformans* viability [40]. Seven distinct bacterial species were isolated including *Bacillus subtilis*, *Escherichia coli*, *Klebsiella aerogenes*, *Proteus mirabilis*, *Pseudomonas aeruginosa*, *Staphylococcus albus*, and *Streptococcus faecalis*. Strikingly, a mixture of these seven bacteria completely inhibited growth of a suspension of *C. neoformans* cells (Figure 2a) [40]. These results indicated that the bacterial mixture had potent anticryptococcal activity. Therefore, it is tempting to speculate that, in addition to the elevated avian body temperature, a specialized avian microbiota may protect birds from infections by *C. neoformans*. Indeed, a study in the early 1980s re-analyzed most of the above mentioned bacteria in individual bacterium–cryptococcal co-cultures, and found that the growth-inhibiting activity was mainly exerted by *P. aeruginosa* and *B. subtilis* [41].

Already in the mid 1950s, it was observed that cultures of the ubiquitous soil bacterium *P. aeruginosa* had the capacity to inhibit the growth of *C. neoformans* by an unknown mechanism [42]. A follow-up study in the mid 1970s analyzed 44 different *P. aeruginosa* clinical strains for their individual potential to inhibit 14 different clinical *C. neoformans* strains [43]. Strikingly, all pseudomonal strains inhibited growth of all 14 *C. neoformans* strains on solid media, although with varying efficiency. The authors noted that the more mucoid fungal strains had a tendency to display increased sensitivity towards *P. aeruginosa*-mediated inhibition [43]. Usually, mucoidy in *C. neoformans* is indicative of normal polysaccharide capsule formation, while strains with a dry colony appearance have reduced capsules [44]. This observation raises the possibility that *P. aeruginosa* may preferentially target encapsulated cryptococcal cells. Although unsuccessful in isolating the precise bacterial factor(s) responsible for the antifungal activity, the authors ruled out an involvement of the pseudomonal phenazine, pyocyanin [43]. Contrary to the findings in the study from 1975 [43], newer studies of the interaction between *P. aeruginosa* and *C. neoformans* indicate that direct bacterial–fungal cell-to-cell-contact triggers the production of pyocyanin and other factors to inhibit cryptococcal growth (Figure 2b) [45]. The differences in both studies regarding the role of pyocyanin may have been due to differences in the growth media used for cultivation. Indeed, it was shown that the inhibitory activity towards *C. neoformans* was fungicidal, and that it was dependent on the pseudomonal cell density and relative ratio of fungal and bacterial cells [45]. Since exogenously supplied pyocyanin only had fungistatic effects on *C. neoformans* cultures, it was concluded that additional bacterial factors, for example proteases and rhamnolipids, might also contribute to the antifungal activity [45]. Notably, *C. neoformans* did not impact the growth of *P. aeruginosa*, indicating that the bacteria have antagonistic activity, while the fungus remains neutral during this interaction [45].

The bacterial pathogen *Staphylococcus aureus* can cause life-threatening infections in humans. Similar to *P. aeruginosa*, *S. aureus* also displays fungicidal activity towards *C. neoformans* during fungal–bacterial co-culture (Figure 2b) [46]. Again, bacterial growth was not affected by *C. neoformans*, indicating a mono-directional interaction. Interestingly, a control experiment also revealed that *C. albicans* growth and survival was not affected by *S. aureus* [46]. This points to the possibility of *Cryptococcus*-specific proteins or factors targeted by the bacterium. Indeed, *S. aureus* cells were observed to preferentially attach to *C. neoformans* cells that have the capacity to form capsule (Figure 2c). Bacteria did not attach to an acapsular mutant of *C. neoformans* [46]. However, since the experiments were not performed under robust capsule-inducing conditions, it would be interesting to study the interaction of *S. aureus* and *C. neoformans* under conditions that promote capsule biosynthesis. Nevertheless, the likelihood that *S. aureus* attaches to capsule polysaccharide is quite high because exogenously added capsular polysaccharides reduced bacterial binding to and killing of *C. neoformans* [46].

Recently, the nosocomial bacterial pathogen *Acinetobacter baumanii* was demonstrated to induce cryptococcal capsule and biofilm formation during co-cultivation (Figure 2c) [47]. The exact molecular mechanism of this interaction remains to be determined, however, the authors established that physical contact was not required, at least for the biofilm-inducing activity. This indicates that *A. baumanii* likely secretes specific factors that affect the fungus either at the cell surface or inside the cell. The interaction between both organisms also resulted in reciprocal killing. Co-incubation experiments revealed that 40–75% of cryptococcal cells were killed by *A. baumanii*, while ~65% of bacterial cells were killed by *C. neoformans* [47]. As noted by other researchers however, *A. baumanii* is not a common soil bacterium and it is unclear whether the observed effects have clinical or biological significance [48].

We recently showed that the ubiquitous soil bacterium *Bacillus safensis* has potent anti-capsular activity, in part via the action of chitinase activity upon cell-to-cell contact (Figure 2c) [49]. *B. safensis* is a Gram-positive, spore-forming bacterium that was first isolated from a Spacecraft Assembly Facility at the Jet Propulsion Laboratory, USA, and it obtained its name from this location (SAFensis) [50]. *B. safensis* was also demonstrated to inhibit melanin formation by *C. neoformans* in a process that relied in part on chitinase activity (Figure 2d) [49]. Physical contact was required for the anti-virulence factor activities, and we hypothesized that the bacterial cell may produce cell surface-associated chitinase(s) upon contact with the fungus, or that contact may trigger close-range secretion of the enzyme [51]. Bacterial proteases and lipopeptides could be other factors involved during the interaction of *B. safensis* with *C. neoformans*. *B. safensis* specifically inhibited cryptococcal virulence factor production without significantly affecting overall fungal growth, thus, it is tempting to speculate that this bacterium or similar *Bacilli* may have the potential of being developed into antifungal probiotics that exclusively target virulence factor production by *C. neoformans* [52,53]. Encouragingly, some *Bacillus* spp., including *B. subtilis* and *Bacillus pumilus*, the latter being closely related to *B. safensis*, have recently been demonstrated to have potent anti-pathogen activities and are already being used as probiotics in certain countries [54,55,56,57].

While *B. safensis* inhibits cryptococcal melanin production, the opportunistic bacterial pathogen *K. aerogenes* was shown to promote melanization of *C. neoformans* cells during co-cultivation (Figure 2d) [58]. The basis for the activity was the bacterial production of dopamine that can serve as a precursor for cryptococcal melanin biosynthesis [58]. Another study established that bacterial homogentisic acid, which is an intermediate product of tyrosine and phenylalanine catabolism, can serve as a precursor for melanin formation by bacteria and *C. neoformans* (Figure 2d) [59].

It was recently shown that the murine microbiota has the capacity to induce titan cell formation by *C. neoformans* (Figure 2e) [60]. Titan cells are cryptococcal cells with enormous dimensions and clinical relevance due to being refractory to phagocytosis by human immune cells [61,62]. The in vivo significance of the microbiota in promoting titan cell formation was established by the finding that mice pre-treated with antibiotics prior to infection with *C. neoformans* had significantly less fungal cells with the titan morphology compared to antibiotic-free mice [60]. Further analysis of the titan cell-inducing mechanisms revealed that bacteria such as *E. coli*, and *Streptococcus pneumoniae* trigger cryptococcal titanization via shedding of peptidoglycan, a component of the bacterial cell wall (Figure 2e) [60].

In summary, these studies demonstrate that different bacteria can have disparate effects on *C. neoformans*, either promoting or preventing growth and survival, and either enhancing or blocking production of virulence factors.

## 4. Direct Cell-To-Cell Contact during Fungal–Bacterial Encounters

Direct cell-to-cell contact is known to be important for interactions among bacteria. For example, *E. coli* has a contact-dependent inhibition system to prevent the growth of competing bacteria [63]. Contact-dependent interactions between bacteria and fungi are far less well understood. Studies on *C. albicans*–bacteria co-incubations revealed that several bacteria have the capacity to attach to the fungal cells via interaction with the fungal surface-localized, hypha-specific and agglutinin-like protein Als3 [64,65]. Accordingly, *P. aeruginosa* only attaches to and kills *C. albicans* hyphal cells. Yeast cells of *C. albicans* are not affected by the bacterium [66].

Bacterium-driven interactions that do not require direct cell contact are usually based on the secretion of specific bacterial molecules that can enter the fungal cell on their own (Figure 3a I). In contrast, there are several ways that direct cell-to-cell contact may trigger an interaction. First, bacterial attachment to the fungus may trigger the injection of factors into the fungus (Figure 3a II). Second, bacterial factors may be produced and impact the fungus at its surface following attachment of bacterial cells (Figure 3a III). Finally, bacteria may attach to the fungal cell surface and form aggregates and biofilms (Figure 3a IV). We have recently shown that the soil bacterium *B. safensis* forms aggregates on some *C. neoformans* yeast cells during co-cultivation (Figure 3b) [49]. Moreover, we found that *B. safensis* can attach to *C. albicans* hypha (Figure 3c) [49]. Interestingly, our unpublished results indicate that *B. safensis* can also attach to cells of the plant-pathogenic fungus *Ustilago maydis* (Figure 3d). Therefore, *B. safensis* has the capacity to attach to diverse fungi. These results also point to the possibility that the fungal molecule(s) that the bacteria use for docking may be conserved among these different fungi. Direct cell contact between bacteria and pathogenic fungi was also observed for the interaction of *C. neoformans* with *P. aeruginosa* and *S. aureus* [45,46].

In summary, it appears that direct cell-to-cell contact between bacteria and fungi is common during interactions, and understanding mechanisms of attachment and bacterial factors that are delivered could potentially lead to the identification of novel antifungal activities.

## 5. Open Questions for Further Study

Despite recent insights into some specific *C. neoformans*–bacterial interactions, many questions remain to be investigated. Specifically, it will be crucial to identify the molecular mechanisms underlying the different types of interactions between *C. neoformans* and bacteria. Some of the open questions are as follows:Which types of bacteria interact with *C. neoformans* in the environment and in the host?How prevalent are cryptococcal–bacterial interactions in nature and in the human host?Do polymicrobial interactions between *C. neoformans* and bacteria lead to emergent properties?Do bacteria impact cryptococcal gene expression?What effect does *C. neoformans* have on bacterial interaction partners?What are the immunological implications of *C. neoformans*–bacteria interactions?Do interactions, e.g., with the microbiota, impact the clinical outcome of cryptococcosis?What are the detailed molecular mechanisms in play during polymicrobial interactions?How may the bacterial factors mediating interaction with *C. neoformans* be identified?How do *C. neoformans*-bacteria interactions evolve?Will it be possible to use certain bacteria as probiotics to prevent or treat cryptococcosis?

In order to answer these questions, it will be important to carefully increase the level of sophistication of fungal-bacterial interaction studies. First, these interactions should be studied in vitro to potentially uncover mechanisms-of-action, and then the complexity could be increased by including interactions in the presence of human cell lines (e.g., lung epithelial cells). For a long time, the human lung has been thought to be a sterile organ. Recent studies however suggest that the lungs have a distinct microbiota [67,68]. In this context, it is intriguing to consider the possibility that the lung microbiota may influence the initial pulmonary infection with *C. neoformans*. Furthermore, fungal–bacterial interactions may be studied in vivo in animal models leading potentially to the discovery of novel probiotic bacteria that antagonize the growth or virulence of *C. neoformans*. Clinical trials with new probiotics may then represent an important translational outcome. Finally, it should be kept in mind that, while bi-microbial interaction studies provide a phenomenal opportunity to uncover potentially new antifungal treatment strategies, ultimately the challenge will be understanding fungal virulence within the complete consortia of thousands of niche-specific microbes. The establishment of model polymicrobial communities may help in studying such interactions [69,70,71].

## 6. Conclusions

The era of multi-species interaction studies has just begun. New discoveries involving the human microbiota are made on an almost daily basis, and many of them have potentially huge clinical implications. Most current research efforts are focused on the impact of the bacteriome on the host, and it will therefore be important to include fungi into future analyses. Fungal pathogens such as *C. neoformans* can cause debilitating infections in humans. Therefore, finding new approaches to tackle these neglected infections is extremely important, and the study of fungal–bacterial interactions may open up the way to discover novel antifungal drug targets and new antifungal compounds.

## Figures and Tables

**Figure 1 jof-05-00031-f001:**
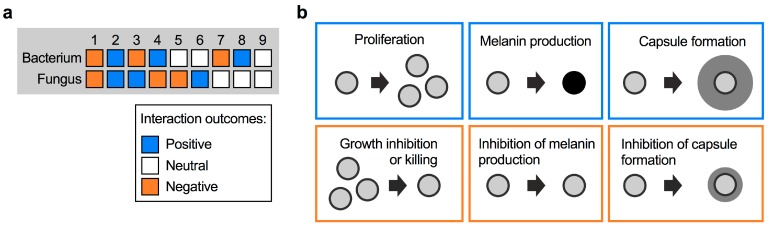
Types of fungal–bacterial interactions. (**a**) Possible types of interactions between fungi and bacteria, and their respective outcomes. (**b**) Examples of positive and negative interaction outcomes for *C. neoformans* cells following exposure to bacteria. Bacteria may induce fungal proliferation, or kill fungal cells. Bacteria may also trigger the expression of fungal virulence factors (e.g., formation of melanin pigment or polysaccharide capsule), or repress formation of these factors. The different outcomes are color-coded depending on their impact on the fungus, i.e., outcomes likely to be beneficial to the fungus are boxed in blue, while outcomes likely to be unfavorable are boxed in orange.

**Figure 2 jof-05-00031-f002:**
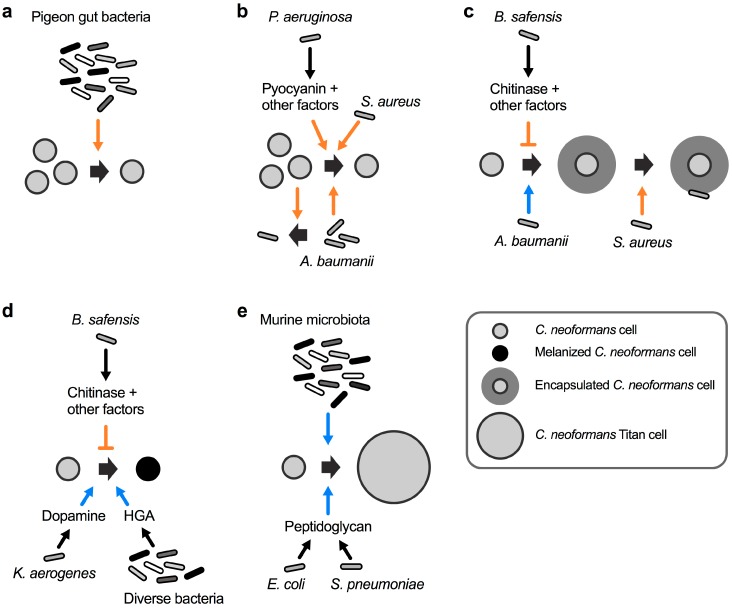
The spectrum of interactions between *C. neoformans* and bacteria. (**a**) A mixture of bacteria isolated from the gastrointestinal tract of healthy pigeons kills *C. neoformans*. (**b**) *Acinetobacter baumanii* and *C. neoformans* reciprocally inhibit each other’s growth. *Staphylococcus aureus* kills *C. neoformans* by an unknown mechanism, and *P. aeruginosa* kills cryptococcal cells via production of pyocyanin and other factors. (**c**) *A. baumanii* induces *C. neoformans* capsule formation, and *S. aureus* preferentially attaches to and kills encapsulated *C. neoformans* cells. *Bacillus safensis* inhibits capsule formation via production of chitinase and other factors. (**d**) *K. aerogenes* produces dopamine, and diverse bacteria produce homogentisic acid (HGA), both of which serve as substrates for cryptococcal melanin biosynthesis. *B. safensis* inhibits fungal melanin production via chitinase activity and other factors. (**e**) Cell wall peptidoglycan from *E. coli* and *Streptococcus pneumoniae* induce *C. neoformans* titan cell formation. The murine microbiota induces fungal titanization by an unknown mechanism. The orange colored arrow-headed and blunt-ended lines indicate inducing and repressive processes, respectively, that have a negative impact on cryptococcal viability or virulence factor production. The blue colored arrow-headed lines indicate processes that have a positive influence on the formation of *C. neoformans* virulence factors.

**Figure 3 jof-05-00031-f003:**
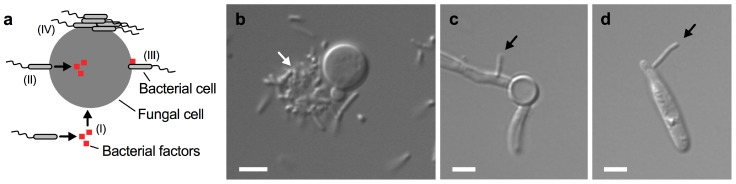
Close-contact interactions of *B. safensis* with diverse fungal pathogens. (**a**) Schematic representation of some fungal-bacterial interactions. (I) Bacteria may secrete certain factors that enter the fungal cell; (II) bacteria may attach to the fungal cell surface and inject factors into the fungus; (III) bacteria may attach to the fungal cell and express cell-surface associated factors; and (IV) bacteria may attach to the fungal cell surface and form cell aggregates and biofilms. Additional mechanisms may exist. (**b**) Differential interference contrast (DIC) microscopy image of *C. neoformans* cells grown with *B. safensis* in yeast peptone dextrose medium for 24 h. Note that *B. safensis* appears to form a cluster of cells (indicated by a white arrow) on one side of the *C. neoformans* cell. (**c**) DIC microscopy image of *C. albicans* cells grown with *B. safensis* under fungal hypha-inducing conditions for 4 h. A bacterial cell (black arrow) can be seen attached to the fungal filament. (**d**) DIC microscopy image of *U. maydis* cells grown with *B. safensis* in potato dextrose broth for 24 h. A bacterial cell (black arrow) has attached to the fungal cell. Scale bars, 2 µm.

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
