# Peer review of "The Spectrum of Interactions between Cryptococcus neoformans and Bacteria"

_jof, 2019, doi:10.3390/jof5020031_

Round 1

Reviewer 1 Report

This is a very nice overview of a highly exciting area of research, in which it summarizes the current studies on the influence of environmental bacteria on the development and virulence of the fungal pathogen Cryptococcus neoformans. Overall, the review article is well written, clear and concise. One potential addition is that how to identify the responsive bacterial factors for the interaction should be one interesting area for further study. Another minor comment is that the high body temperature in birds has been proposed as an inhibitory factor for C. neoformans proliferation. Authors may want to mention this in their discussion of the bacterial factors. 

Author Response

Response: We thank the reviewer for the positive comments and the useful suggestions.  We added a point about identifying bacterial factors as suggested, and we also added a phrase about the high body temperature of birds.

Reviewer 2 Report

An excellent review by Mayer and Kronstad, covering a very important research area.  I only have minor text corrections to suggest:

Abstract Line 11: change "and" for "or" in the sentence "either in the environment and in the human host.

Introduction lines 22-25: These two sentences repeat the same point. Please form one sentence here.

Introduction line 29 and 33: use of "main"  is informal/non-specific. Suggest predominant, common etc

Section 2 lines 86-89: need to make it clear that you are postulating at this point ”could have” or “ One could imagine multiple outcomes” etc. 

Section 3 Line 118: insert "the" or change to "by the" in the sentence starting "Already in mid 1950s..."

Section 3 Line 158: delete "major"

Section 4 Lines 235-236: red line in text between sentences.

These minor changes are also highlighted in the attached pdf file

Author Response

Reviewer 2.  An excellent review by Mayer and Kronstad, covering a very important research area.  I only have minor text corrections to suggest:

Abstract Line 11: change "and" for "or" in the sentence "either in the environment and in the human host.

Response: Corrected

Introduction lines 22-25: These two sentences repeat the same point. Please form one sentence here.

Response:  The reviewer makes a good point about the two sentences.  We reviewed the text and removed the phrase” as well as a variety of organisms in other kingdoms”  to eliminate the redundancy between the two sentences.

Introduction line 29 and 33: use of "main"  is informal/non-specific. Suggest predominant, common etc

Response: Corrected as suggested

Section 2 lines 86-89: need to make it clear that you are postulating at this point ”could have” or “ One could imagine multiple outcomes” etc. 

Response: Corrected as suggested

Section 3 Line 118: insert "the" or change to "by the" in the sentence starting "Already in mid 1950s..."

Response: Corrected as suggested

Section 3 Line 158: delete "major"

Response: Corrected

Section 4 Lines 235-236: red line in text between sentences.

Response: Corrected